# A Product Conceptual Design Method Based on Evolutionary Game

**Yun-Liang Huo [1], Xiao-Bing Hu [1],\*, Bo-Yang Chen [1] and Ru-Gu Fan [2]**

[1]  School of Science and Engineering, Sichuan University, Sichuan 610065, China;
     huo_yunlliang@163.com (Y.-L.H.); shanbenyunque8018@163.com (B.-Y.C.)
[2]  Sinohydro Jiajiang Hydraulic Machinery Company Limited, Sichuan 614000, China; liofrg@163.com
\*   Correspondence: huxb@scu.edu.cn

**Abstract:** In this paper, an intelligent-design method to deal with conceptual optimization is proposed for the decisive impact of the concept on the product-development cycle cost and performance. On the basis of matter-element analysis, an effective functional-structure combination model satisfying multiple constraints is first established, which maps the product characteristics obtained by expert research and customer-requirements analysis of the function and structure domain. Then, the Evolutionary Game Algorithm (EGA) was utilized to solve the model, in which a strategy-combination space is mapped to the solution-search space of the conceptual-solution problem, and the game-utility function is mapped to the objective functions of concept evaluation. Constant disturbance and Best-Response Correspondence were applied cross-repeatedly until the optimal equilibrium Pareto state corresponding to the global optimal solution was obtained. Finally, the method was simulated on MATLAB 8.3 and applied to the design for fixed winch hoist, which greatly shortens its design cycle.

**Keywords:** conceptual design; intelligent design; evolutionary game; domain mapping

## 1. Introduction

Research on product conceptual design is booming with regard to the direct influence of the concept on the quality of the final product, and the vast majority of researchers agree that how to scientifically evaluate candidate concepts and how to express a product concept with an accurate model are two vital tasks in conceptual design [1]. Hence, advanced models and effective evaluation systems have been intensively addressed by researchers worldwide. Danni et al. [2] presented an evaluation and selection method composed of three modules: data mining, concept reconstruction, and decision support, to improve the efficiency of concept review and evaluation. Sun et al. [3] established an effective conceptual model for new-product concept development from two theoretical backgrounds about organizational learning, and the model was applied to the design of a large scramjet with satisfying results. Wang et al. [4] proposed an optimization decision model for product conceptual design to help enterprises select key technical characteristics under the condition that cost and time maximally meet customer requirements. Christoph F. et al. [5] presented methodology integration with a knowledge model for conceptual design in accordance with model-driven engineering, and the work extended Gero's Function-Behavior-Structure model. Based on Bunge's Scientific Ontology, Chen et al. [6] developed an explicit and complete conceptual foundation for the establishment of a new conceptual design model. Varun Tiwari et al. [7] proposed a novel way of performing design-concept evaluations, where instead of considering the cost and benefit characteristics of the design criteria, the work identifies the best concept that satisfies constraints imposed by the team of designers, as well as fulfilling as many of a customer's preferences as possible. To obtain the best comprehensive

performance of mechanical products, Wang et al. [8] established an evaluation model for product conceptual design based on the principle of maximum-entropy value, and solved the model by constructing a Lagrange function.

The above work mainly focuses on product-model expression and product conceptual evaluation. However, there could be many generated concepts through its combination nature, and the evaluation of a larger number of concepts, one by one, is a very difficult work, although many novel and effective methods of concept evaluation have been proposed [6–8]. As a result, the best design concept cannot easily be obtained, and the internalization of the conceptual-design process becomes critical.

Computational intelligence, which consists of an evolutionary neural network and fuzzy logic, is a novel technology aiming to bring intelligence into computation [9]. Attempts have been made in recent years for the application of computational intelligence. Manu Augustin [10] proposed a framework that uses a fuzzy inference process for evaluating each initial concept against identified decision criteria, to select and/or evolve improved concepts. Integrated with ACO, Ma et al. [11] presented a mathematical programming model to quantitatively predict change-propagation impact, and improved the intelligence of change-propagation prediction during the design process. Ming-Chyuan et al. [12] proposed an integrated procedure that involves neural-network training and genetic-algorithm simulations within the Taguchi quality-design process to aid in searching for an optimal solution with more precise design-parameter values for improving product development. Oliviu Matei et al. [13] addressed the automated product-design problem with two distinct evolutionary approaches: genetic algorithms and evolutionary ontology. S.H. Ling [14] developed intelligent particle-swarm optimization (iPSO), where a fuzzy-logic system, developed based on human knowledge, is proposed to determine the inertia weight for the swarm movement of the PSO and the control parameter of a newly introduced cross-mutated operation.

Although the above methods greatly contribute to the process of conceptual-design intelligence, the main focus is to study the commonality of various problem models [4,6,13,14]. While the model can be solved to obtain a feasible solution, they ignore the personality of the problem. If we choose or design a specific algorithm to solve a specific problem, the efficiency and accuracy of the solution is improved [8]. In view of this, we explored the establishment of a constraint model for product design, focusing on the functional variables and constraints of the model, and the optimal or approximately optimal solution of the functional variable combination of the Evolutionary Game Algorithm (EGA) search model was completed in this paper. In order to accurately express information during conceptual design, product characteristics are extracted at first via customer-requirement analysis and the application of expert knowledge, and the Analytic Network Process (ANP) is used to assess their importance. Then, a model of product conceptual design is established by means of mapping product characteristics to the functional and structural domains while comprehensively taking all constraints of product conceptual design into account. Finally, to quickly solve the model, intelligent algorithm EGA, with fast convergence speed, was used [15], and the optimal solution was obtained after multiple evolutions.

The paper is organized as follows. Section 1 introduces the process of how a product-optimization design model is established. Section 2 briefly introduces EGA. Section 3 provides a practical example to illustrate how the method performs. Section 4 concludes the paper.

## 2. Modeling for Product Conceptual Design

### 2.1. Matter-Element Description

The matter-element model is a representation of objects for computer storage, recognition, and operation, which is widely employed in product design and reliability assessment. Yue et al. [16] applied matter-element theory to ecological-risk assessment, and successfully evaluated the Gannan Plateau. To solve the formal description in the modular design of mechanical products, Huang et al. [17] introduced extension theory into Reconfiguration Design Technology (RDT), and

built the matter-element model. Based on the model, the selection, matching, and transformation of a mechanical product and its modules were researched. Liu [18] proposed an assessment approach by combining extension and ensemble empirical-mode decomposition (EEMD) to describe the bearing performance-degradation (BPD) process that was denoted by the matter-element model. Lv et al. [19] presented a new method for equipment-criticality evaluation based on a fuzzy matter-element model.

In this paper, a model of product conceptual design based on matter-element analysis was constructed. Firstly, the function tree and structure tree could be obtained by mapping product characteristics to functional and structural domains, before which the product characteristics and their importance must be obtained through expert investigation and customer-requirement analysis. Then, in order to obtain the utility function of a product, various constraints in conceptual design are comprehensively considered, and the utility vector of the product characteristics is given to each substructure with the knowledge of the expert team. Finally, a matter-element model of product conceptual design is established with both utility attributes and design-constraint attributes invested to express product information.

The above model can be described as *Pro* = (*S_Attrib*, *U_Attrib*, *C_Attrib*, *Cl_Attrib*), where *U_Attrib* denotes the model-evaluation information of each product concept; *Cl_Attrib* denotes the hierarchical information of the matter element; *C_Attrib* denotes the constraint information including functional, structural, and relational constraints during product conceptual design; and *S_Attrib* denotes product-feature information. In order to express it more clearly, the matter-element model of product conceptual optimization design is expressed as follows:

$$
\begin{pmatrix}
Pro, & U\_Attrib, & v_1 \\
 & Cl\_Attrib, & v_2 \\
 & C\_Attrib, & v_3 \\
 & S\_Attrib, & v_4
\end{pmatrix}
\tag{1}
$$

where $v_i$ ($i$ = 1, 2, 3, 4) is the value of an attribute belonging to a matter element, the larger the $v_1$, the better the concept; $v_2$ denotes the hierarchical information of the matter element; $v_3$ indicates whether the solution is a feasible solution, for example, $v_3 = 0$ means that the solution is feasible without breaking any constraint; and $v_4$ is the combined information of the substructures for achieving a functional unit. The detailed information of each attribute is expressed via its submatter elements, and the process of finding the optimal solution is transformed into the process of searching for a matter element of a product concept with a maximum $v_1$ under constraint conditions $v_3$ via combination of submatter elements.

*2.2. Matter-Element Description*

Product-characteristics set *PC* is obtained through the brainstorming of experts and technicians involved in all phases of the product life cycle, with customer requirements being taken into consideration (the $i$th element in *PC* is denoted by $PC_i$). The original *PC* should be processed to obtain the new one, as their relationships may be inclusion, cross, and independence. Generally speaking, there are mutual relations between elements in *PC*, customer-requirement set *CR* (the $j$th element in *CR* is denoted by $CR_j$) and $PC_i$, which should all be taken into account when synthetically analyzing the importance of *PC*. The Analytic Network Process (ANP) method is a widely used decision-making algorithm, mainly to determine the relative importance of a group with inter-related elements in a multiobjective decision-making problem; therefore, it is adopted to analyze a *PC* and calculate its importance.

1. Analyzing the importance of a *PC* driven by *CR*

Assume that each $PC_i$ is independent from the others. Importance vector $w_s = (w_1, w_2, \ldots w_m)$ is obtained according the customers' preference for each requirement. For each $PC_i$, relative importance matrix $R_i$ between *CR* and $PC_i$ is evaluated by an expert team; for element $r_{ij} \in [0,9]$ in $R_k$, which

indicates the importance of $RC_i$ for $RC_j$ when pursuing $PC_k$, if $r_{ij} \neq 0$, then $r_{ji} = 1/r_{ij}$; else, $r_{ij} = r_{ji} = 0$. In addition, the Analytic Hierarchy Process (AHP) [20] is used to obtain relative-importance vector $w_i$ = $(w_{1i}, w_{2i} \ldots w_{ii}, w_{mi})$, where $\sum_{j=1}^{m} w_{ji} = 1$, and importance matrix $W_{cr\text{-}pc}$ of a PC driven by CR can finally be obtained.

$$
R_k = \begin{pmatrix} r_{11} & r_{12} & \cdots & r_{1i} & \cdots & r_{1m} \\ r_{21} & r_{22} & \cdots & r_{2i} & \cdots & r_{2m} \\ \vdots & \vdots & \vdots & \vdots & \cdots & \vdots \\ r_{i1} & r_{i2} & \cdots & r_{ii} & \cdots & r_{im} \\ \vdots & \vdots & \vdots & \vdots & \vdots & \vdots \\ r_{m1} & r_{m2} & \cdots & r_{mi} & \cdots & r_{mm} \end{pmatrix} \quad W_{cr-pc} = \begin{pmatrix} w_{11} & w_{12} & \cdots & w_{1i} & \cdots & w_{1n} \\ w_{21} & w_{22} & \cdots & w_{2i} & \cdots & w_{2n} \\ \vdots & \vdots & \vdots & \vdots & \cdots & \vdots \\ w_{i1} & w_{i2} & \cdots & w_{ii} & \cdots & w_{in} \\ \vdots & \vdots & \vdots & \vdots & \vdots & \vdots \\ w_{m1} & w_{m2} & \cdots & w_{mi} & \cdots & w_{mn} \end{pmatrix} \tag{2}
$$

where $w_{ij}$ denotes the impact degree of $CR_i$ on $PC_j$, and vector $w^{(1)} = w_s \times w_{cr\text{-}pc}$ denotes the importance of a PC driven by CR.

2.　Gaining mutual importance among elements of PC

Relative-importance degree matrix $R'_i$ that is similar to $R_i$ is obtained when considering the correlations between $PC_i$ and the others. $r_{ij}$ in R'i indicates the importance of $PC_i$ for $PC_j$ when pursuing $PC_k$, and importance vector $w^{(2)} = (w_1^{(2)}, w_2^{(2)}, \ldots \ldots w_n^{(2)})$ is also obtained by AHP, where $\sum_{j=1}^{n} w_i^{(2)} = 1$.

$$
R_k^{'} = \begin{pmatrix} r_{11} & r_{12} & \cdots & r_{1i} & \cdots & r_{1n} \\ r_{21} & r_{22} & \cdots & r_{2i} & \cdots & r_{2n} \\ \vdots & \vdots & \vdots & \vdots & \cdots & \vdots \\ r_{i1} & r_{i2} & \cdots & r_{ii} & \cdots & r_{in} \\ \vdots & \vdots & \vdots & \vdots & \vdots & \vdots \\ r_{n1} & r_{n2} & \cdots & r_{ni} & \cdots & r_{nn} \end{pmatrix} \quad W_{pc} = \begin{pmatrix} w_{11} & w_{12} & \cdots & w_{1i} & \cdots & w_{1n} \\ w_{21} & w_{22} & \cdots & w_{2i} & \cdots & w_{2n} \\ \vdots & \vdots & \vdots & \vdots & \cdots & \vdots \\ w_{i1} & w_{i2} & \cdots & w_{ii} & \cdots & w_{in} \\ \vdots & \vdots & \vdots & \vdots & \vdots & \vdots \\ w_{n1} & w_{n2} & \cdots & w_{ni} & \cdots & w_{nn} \end{pmatrix} \tag{3}
$$

$$
w_i^{(2)} = \frac{\sum_{j=1}^{n} w_{ji}}{n} \tag{4}
$$

3.　Gaining the importance of PC

The importance of $PC_i$ is shown in Equation (5) by comprehensively considering the two relationships mentioned above.

$$
w_i = \frac{w_i^{(1)} \times w_i^{(2)}}{\sum_{k=1}^{n} w_k^{(1)} \times w_k^{(2)}} \tag{5}
$$

*2.3. Modeling Process*

2.3.1. Multidomain *PC* mapping

With the fuzzy, complex, and tedious relationships between *PC* and the product structure, inaccuracy of information mapping and loss of information occur if we directly map the *PC* to the product-structure domain. Therefore, considering the correspondence between product function and structure in axiomatic design [21], the functional domain is introduced as an intermediate medium between *PC* and product structure, guiding mapping the *PC* to the product domain, and completing the product-structure design of the specific $PC_i$.

### 2.3.2. Function Decomposition

The process of *PC* multidomain mapping is shown in Figure 1, where the product-function tree is obtained by progressively decomposing product function to the tiniest independent functional units; the structure tree corresponding to the function tree is obtained by an expert team that enumerates the component structure corresponding to each function in the product-design database; the cell located at the bottom of the structure tree is called a substructure.

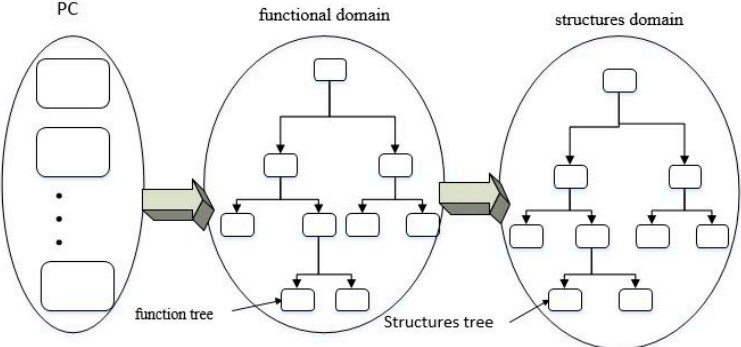

**Figure 1.** Product-characteristics set (PC) multidomain-mapping diagram.

### 2.3.3. Concept Modeling

Both functional units and substructures are denoted by the matter element after finishing the multidomain mapping of *PC*. The optimal-design concept is obtained by solving the model through the EGA via mapping product features to game players. A mechanical-product concept is expressed in Figure 2; it has *n* functional units, and *i*th functional units have *k* substructures.

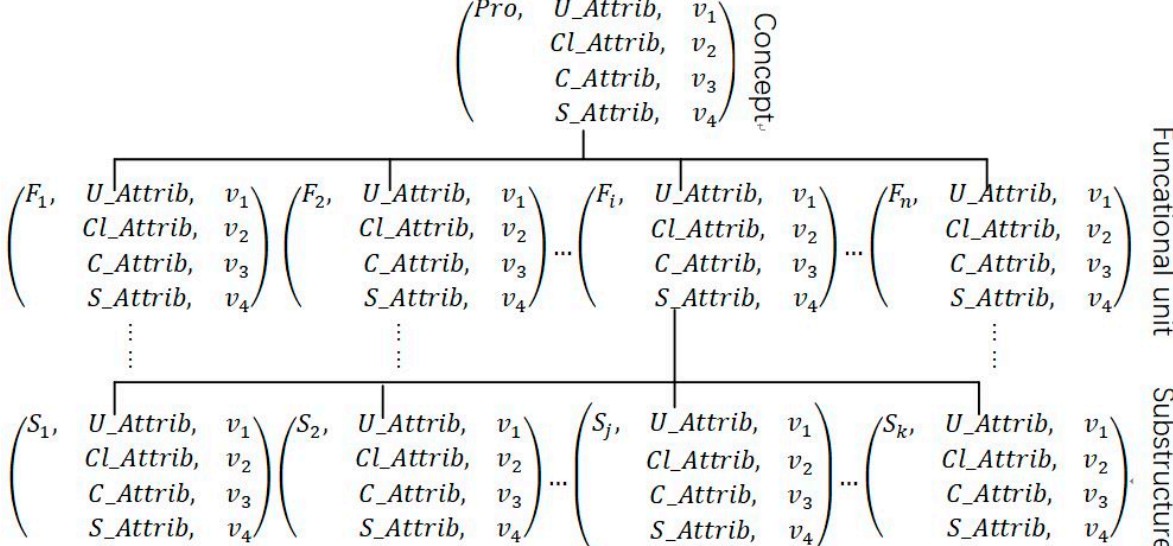

**Figure 2.** Matter-element model of a mechanical product.

where the information of the entire product concept is denoted by the matter-element model, for example, the specific information of the *i*th functional unit of the product, which includes structural information *S_Attrib*, constraint information *C_Attrib*, utility information *U_Attrib*, and hierarchical information *Cl_Attrib*, is denoted by the second-level matter element. The substructures to achieve a functional unit are denoted by third-level matter elements. It should be noted that the third-substructure-layer matter elements are alternative substructures, which are optional strategies of

the game player, since the effectiveness of structural combinations has not been judged; therefore, no constraint information is required.

### 2.3.4. Values of Obtained Matter-Element Attributes

Linguistic terms such as 'very unimportant' and 'medium' are usually used to assess an attribute's importance, as they are always fuzzy during product design. Some linguistic terms should be transferred to crisp numbers for accurate analysis and calculations.

- Strategy variables and utility vectors are obtained

For $m$ substructures $s_{ij}$ ($j = 1, 2 \dots m$) corresponding to a functional unit $f_i$ ($i = 1, 2 \dots n$), one of them must be chosen to achieve $f_i$ during conceptual design, and the choice information of $f_i$ for $m$ substructures can be donated by the value of $S\_Attrib$. For example, if $m = 8$ and the fourth substructure is chosen, then the value of $S\_Attrib$ of $f_i$ is $v_4 = 00010000$, and the utility vector for the $PC$ of substructure $s_{i4}$ is used to calculate the utility value of the product concept. A typical mapping relationship between product features and matter-element attribute values is expressed in Figure 3.

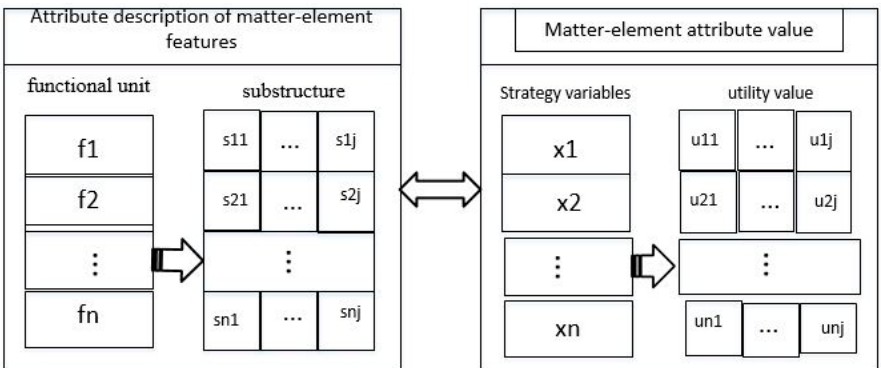

**Figure 3.** Schematic of property values.

where $u_{ij}$ is a utility vector of a $j$th substructure of an $i$th functional unit provided by experts and designers based on a nine-point scale [22], which denotes the utility index of $s_{ij}$; $x_i$ is the strategy variable of functional unit $f_i$, which denotes the choice information of alternative substructures. In the matter-element model proposed above, $x_i$ is the value of $S\_Attrib$ for a game player. The frequently used nine-point scale is shown in Figure 4.

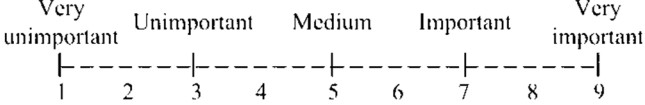

**Figure 4.** Nine-point scale.

- $C\_Attrib$ value is obtained

Product-design constraints, including functional, structural, and related constraints are ultimately embodied in the portfolio optimization of product substructures. In the optimization model proposed in this paper, a uniform expression $C$ was used to specify dependency constraints that can denote the multiple constraint forms, and a dual constraint is taken as an example, shown in Equation (4).

$$C\left(x_i, x_j\right) = \left\{\left(u_{ik}, u_{jp}\right)\right\} \tag{6}$$

where $x_i$ and $x_j$ are variables denoting the constraint relationship between functional units $i$ and $j$, and the ranges of $x_i$ and $x_j$ are expressed as $u_{ik}$ and $u_{jp}$, respectively. $C$ indicates that $j$ must choose the $p$th substructure if $i$ chooses the $k$th. The constraint between a fixed winch hoist coupling and its service brake is used as an example.

$$C\ (b,\ c) = \{(\text{Wheel break, Wheel coupling}),\ (\text{Disc break, Disc coupling})\} \qquad (7)$$

This shows that the wheel brake must be matched with the wheel coupling; otherwise, the number of constraints on the current composition strategy increases. If the number of constraints in the current combination strategy is $i$, then the value of $C\_Attrib$ $v_3$, which is used to decrease the utility value of a concept in an evolutionary game, is $i$.

- *Cl_Attrib* value is obtained

The *Cl_Attrib* attribute in the model mainly denotes the hierarchical information of the matter element. As shown in Figure 2, $v_2 = 1$ indicates it is just a matter element of the product concept rather than a component.

*2.4. Benefits*

●By focusing on functional variables and constraints of the model, the obtained solution is the optimal solution that satisfies the constraint.

●Comprehensively considering PCs and CRs makes products perform well in terms of performance and personalization.

●A modular product functions as a player in the EGA that performs well on combinatorial optimization problems, and quickly obtains the optimal solution.

## 3. Introduction of Evolutionary-Game Algorithm

Considering that product conceptual design is actually a combinatorial optimization problem, EGA was employed to solve the above optimization model as it is effective in solving combinatorial optimization problems [23]. The optimal solution is obtained through the game for functional-unit-layer matter elements, a combination of substructure-layer matter elements, and comparison between matter elements in the conceptual layer.

The EGA is a novel kind of intelligent computation algorithm based on economic game theory and dynamic evolution calculation, which takes maximum utility as its optimization objective and searches the whole solution space by combining the strategies of game players, and simultaneously considers local and global performances. Compared with the selection process of a stochastic genetic algorithm, the EGA converges to a global optimal solution with probability 1, and is more certain in evolution [24].

*3.1. Key Issues*

3.1.1. Fundamental Theorems

A basic game consists of game player $i$, strategy set $S$, and utility $u$; the two fundamental theorems for EGA are shown as follows.

- If strategy combination $S^*$ satisfies Equation (8) for any strategy $s_i \in S_i$ of any game player $i$, then it is called an $S^*$ Nash equilibrium, and $S_i$ is the strategy set of $i$. The specific form of Equation (8) is as follows:

$$u_i\left(s_i^*, S_{-i}^*\right) \geq u_i(s_i, S_{-i}) \qquad (8)$$

where $S_{-i}$ is the strategy combination of players without $i$, $S_{-i}^{\ *}$ is the Nash equilibrium of strategy combinations of players without $i$, and $s_i^*$ is the optimal strategy for $i$ in a Nash equilibrium. It is called a strict Nash equilibrium when

$$u_i\left(s_i^*, S_{-i}^*\right) = u_i(s_i, S_{-i}) \qquad (9)$$

- Assuming that $S_{-i} = \prod S_k$, where $k = 1,2 \ldots n$ and $k \neq i$. If Equation (10), established as follows, is satisfied, then $B_i$ is called the Best-Response Correspondence for player $i$.

$$B_i(s_{-i}) = \{s_i^* \in S_i : u_i(s_i^*, S_{-i}^*) \geq u_i(s_i, S_{-i}), \forall s^{(i)} \in S_i\} \tag{10}$$

Underlying the meaning of the Best-Response Correspondence is a process where $i$ chooses the strategy with the maximum utility in the current situation. The dynamic process that all game players complete a Best-Response Correspondence in turn is called the Optimal-Response Dynamic.

### 3.1.2. EGA Expression

The specific form of the evolutionary-game algorithm is expressed as EGA= {G, S$_0$, $\alpha$, $\beta$, $\tau$}, and each member of the EGA is described in detail as follows.

- Game structure $G$

The game structure is described as $G = [I, S, U]$, where $I$, $S$, and $U$ denote the information of the game players, the current situation, and utility, respectively. For the model described in this work, game-player set I is obtained by mapping functional units to the strategy variables, and $k$ substructures for realizing a functional unit are mapped to the strategy set of the player.

The mathematical description is $s_{ij} \in \{0,1\}$, where $i \in I$ ($1 \leq i \leq n$) and $1 \leq j \leq k$; for example, if $k = 7$ and the third substructure is selected when he functional unit $i$ generates a strategy, according to Section 2.3.3, the strategy of player $i$ transfers to binary code 0010000. Then, the strategy combination of $n$ players constitutes a solution $S$ (also called a situation) in the above model. Equation 11 is the form of utility function $f(S)$ that is used to calculate utility value $U$ of the current situation.

$$U_i = \begin{cases} f(S) & if\ satisfy\ the\ constraint \\ f(S) - f_{max} & else \end{cases} i \in I$$

$$f(S) = \sum_{i=1}^{n} G_i W_i \tag{11}$$

$$G_i = \sum_{k=1}^{m} u_{ij} * W_{PC}$$

where $u_{ij}$ is the utility vector of substructure $s_{ij}$, $m$ is the element number of $PC$; $W_{PC}$ is the importance degree of $m$ $PC_i$ calculated by Equation (5); $W_i$ is the importance degree of game player $i$ in product conceptual design, where $\sum_{i=1}^{n} W_i = 1$; $n$ is the number of game players; and $f_{max}$ is the maximum-utility value of the current evolutionary generation. Compared with penalty functions in other algorithms that are difficult to determine forms, $f_{max}$ can be directly calculated. It should be noted that game player $i$ is only bound by the constraint rules associated with itself during the game.

- Initial situation S$_0$

The EGA starts with S$_0$, which is initialized by a randomization method.

- Optimization operator $\alpha$

Game theory is based on the assumption that all game players are economic, and in the process of evolution, each game player pursues maximum-utility values. Hence, the Best-Response Correspondence is called the optimization operator for the maximum-utility value of a game player made by it.

- Equilibrium perturbation operator $\beta$

In order to ensure that the solution obtained by the EGA is globally optimal, equalization perturbation operator $\beta$ is employed to break the current Nash equilibrium state reached after several iterations; then, a new Nash equilibrium state is obtained by performing the Best-Response

Correspondence of each player after the balance state is sequentially broken. The specific calculation form of $\beta$ is shown in Equation (12):

$$\beta(s_i) = \begin{cases} s_i & if X_i \geq p_i \\ Z_i & else \end{cases} \tag{12}$$

where $p_i$ is the perturbation probability assigned according to the importance degree of each functional unit, and the functional units with much contribution to the utility value of a solution are easier to deviate the system from the original state. Therefore, higher disturbance probability should be attached to them, and the functional units with less contribution to the utility value should be given lower probability. $X_i$ is a decimal randomly generated from 0 to 1; $si$ indicates that the disturbance operator changed nothing and the former strategy is maintained; $Z_i$ is the disturbance operator that means a strategy is randomly selected from the strategy set of player $i$ to replace the current one.

- Termination condition $\tau$

In a given situation, the process of Optimal-Response Dynamic is called one round, and the Nash equilibrium state of the situation is reached after two rounds. Two rounds achieving a Nash equilibrium state are defined as a generation. Setting the iteration termination condition as $\tau \geq T$, and $T$ is the preset iteration generation.

### 3.2. EGA Process

The specific process of EGA is shown in Figure 5.

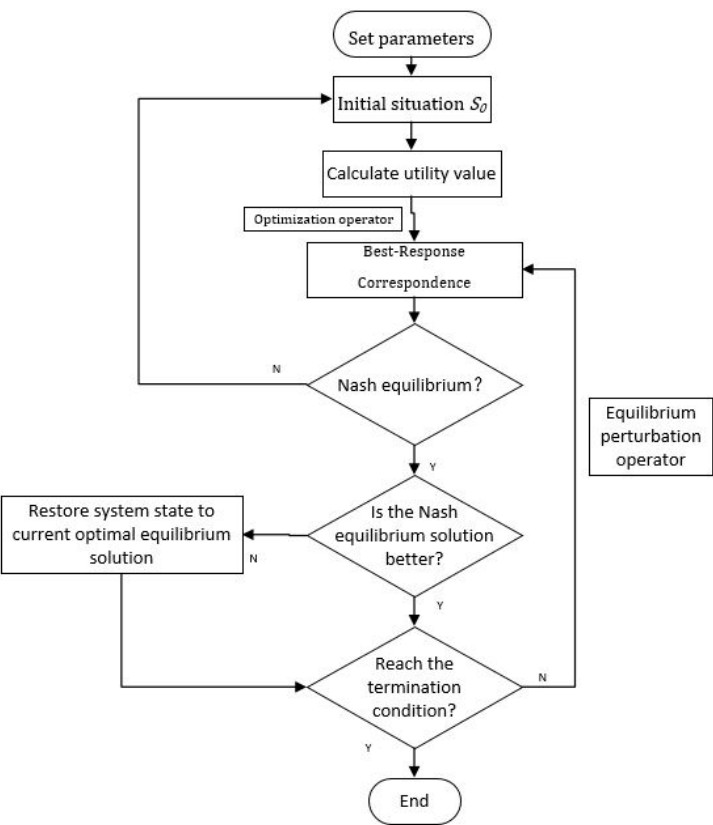

**Figure 5.** Evolutionary Game Algorithm (EGA) flowchart.

**Step 1**: Set parameters.
First, maximum iteration number T and disturbance probability pi are set.
**Step 2**: Algorithm initialization.

Update game structure to $G = G_0$ with the strategy randomly initialized; then, initial situation $S_0$ is generated, and the system starts to evolve from $S_0$ when $\tau = 0$.

**Step 3**: Calculate current-situation utility value.

Calculate the U of the current situation based on *f(S)*.

**Step 4**: Application of optimization operator $\alpha$.

$\alpha$ is first used to estimate updated player utility, and then to update the strategy combination of game players from $S_j$ to $S_{j+1}$ when the updated one is better than the before; otherwise, keep $S_j$ unchanged.

**Step 5**: Stability of the situation.

If the situation at timer $\tau = \tau(i)$ satisfies $U_{i+1} = U_i$, then strategy $S_i$ is stable and its corresponding solution is a Nash equilibrium solution.

**Step 6**: Application of equalization perturbation operator $\beta$.

The new situation is achieved by applying $\beta$ to the current situation; then, update situation $S_j$ to $S_{j+1}$ and calculate the utility value of $S_{j+1}$. Finally, estimate whether it is a stable evolution strategy again.

**Step 7**: Estimation of termination condition.

The algorithm terminates when $\tau \geq T$ is satisfied; otherwise, it returns to Step 6.

The EGA steps can be regarded as a stochastic process in a Nash equilibrium solution space that continuously updates the current stable solution with a better Nash equilibrium until the optimal situation equilibrium is reached. Since the main operation of EGA is only to compare the utility value between different strategy combinations, the global optimal solution can always be obtained by reasonably setting the number of iterations, because the utility of the global optimal solution is greater than other feasible solutions, and the utility of all feasible solutions is greater than infeasible solutions. Compared with frequently used evolutionary algorithms, such as Genetic Algorithm, Ant-Colony Algorithm, and Artificial Neural Networks, which involve complex mutation operations, path calculation, and network learning, respectively, the speed and efficiency of EGA are obvious advantages.

## 4. Case Study

A 3600 KN fixed winch hoist that was supported by the Sinohydro hydraulic machinery company was taken as an example to validate the method mentioned above. *PC* set F for the fixed winch hoist was obtained by product investigation, customer-requirement analysis, technical, economic, and social environments, and, finally, an expert team. Given F = {① low complexity, ② manufacturability, ③ assembly ability, ④ reliability, ⑤ mechanical strength ⑥ environment-friendly, ⑦ brake, ⑧ low noise, ⑨ Lifting stability, ⑩ low cost, ⑪ synchronicity, ⑫ high energy-conversion efficiency, and ⑬ lightweight}. How to express and implement the element-matter model of a fixed winch hoist is introduced below.

### 4.1. Modeling of the Fixed Winch Hoist

#### 4.1.1. Design Knowledge

A fixed winch hoist is a heavy-tonnage lifting machine that works in the water-conservancy and hydropower industries, and it is composed of 10 components. As shown in Figure 6, only the structure of a movable pulley is related to the lifting force, and the structure of the other components could have different structures according to different *PCs*. Hence, the process of conceptual design according to a *PC* is transformed into the process of selecting the optimal structure of each component based on product characteristics.

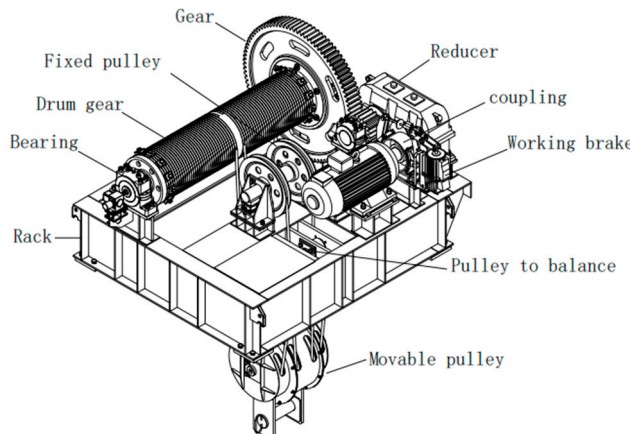

**Figure 6.** Schematic diagram of fixed winch hoist.

In order to solve the problem of fixed-winch-hoist conceptual design from the perspective of product characteristics, the function tree was first obtained by an engineer through functional decomposition, with the substructure set for each functional unit enumerated as shown in Table 1. Then, based on knowledge and customer-requirement constraints, they were obtained as shown in Figure 7. Finally, a model for product conceptual design was established, as shown in Figure 8.

**Table 1.** Function units and their alternative substructure.

| Functional Unit | Structure | Alternative Substructure |
|---|---|---|
| Lifting | $S_1$ Drum gear | $s_{11}$ single helix with intermediate rope, $s_{12}$ single helix with sides rope, $s_{13}$ Single fold with center rope, $s_{14}$ single fold with sides rope $s_{15}$ double fold with intermediate rope, $s_{16}$ double fold with sides rope, $s_{17}$ double helix with sides rope, $s_{18}$ double helix with intermediate rope. |
| Balance | $S_2$ Pulley to balance | $s_{21}$ balanced pulley suspension, $s_{22}$ balanced pulley placement, |
| Stabilization | $S_3$ Fixed pulley | $s_{31}$ fixed pulley is placed vertically, $s_{32}$ fixed pulley is hung vertically, $s_{33}$ fixed pulley is arranged in parallel, $s_{34}$ No fixed pulley |
| Working brake | $S_4$ Working brake | $s_{41}$ wheel brake, $s_{42}$ disc brake |
| Reducer | $S_5$ Reducer | $s_{51}$ horizontal speed reducer, $s_{52}$ suspension reducer |
| Support | $S_6$ Bearing | $s_{61}$ Antifriction bearing, $s_{62}$ sliding bearing, $s_{63}$ hybrid bearing |
| Power transmission | $S_7$ Gear and coupling | $s_{71}$ wheel coupling with gear, $s_{72}$ disc coupling with gear, $s_{73}$ wheel coupling, $s_{74}$ disc coupling |
| Safety brake | $S_8$ Safety brake | $s_{81}$ safety brake, $s_{82}$ no safety brake |
| Master support | $S_9$ Rack | $s_{91}$ motor fixed pulley same side, $s_{92}$ motor fixed pulley different side |

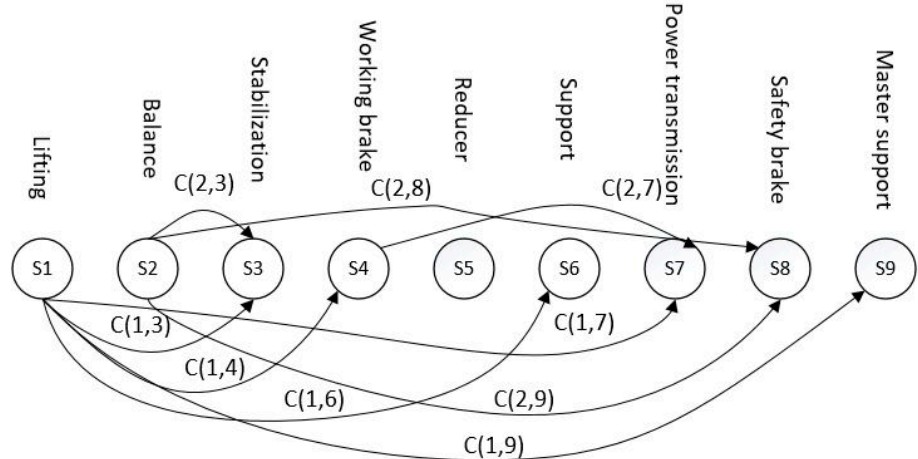

**Figure 7.** Constraint expression of hoist conceptual design.

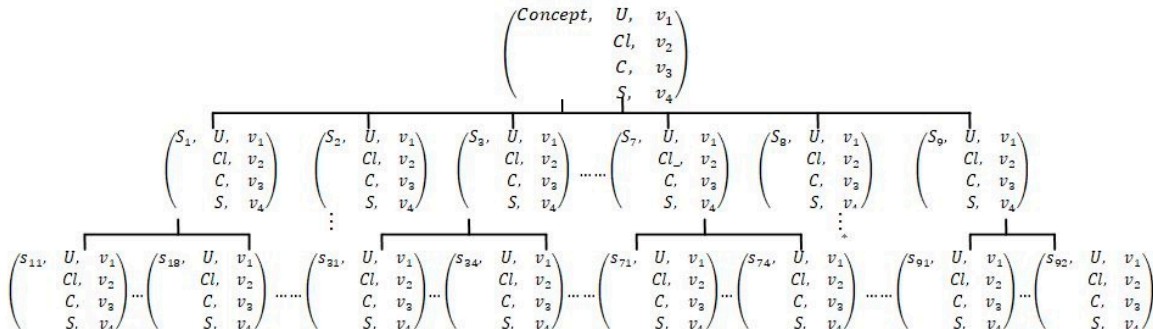

**Figure 8.** Matter-element model of fixed winch hoist.

### 4.1.2. Acquiring *PC* importance

Analyzing the importance of PC driven by CR

According to customer preferences, the weight vector for five customer requirements *CR* = {① maintainability, ② long service life, ③ work stability and reliability, ④ energy utilization rate, ⑤ environment-friendly}, $w_s$ = (0.29, 0.30, 0.31, 0.09, 0.05) appeared, and *PC* importance relationship matrix $W_{cr\text{-}PC}$ driven by customer requirement is obtained using the Analytic Hierarchy Process. Relative importance matrix $R_i$ between *CR* and $PC_i$ is evaluated by an expert team, and $R_1$ was taken as an example

$$CR_1\ CR_2\ CR_3\ CR_4\ CR_5 R1 = \begin{array}{c} CR_1 \\ CR_2 \\ CR_3 \\ CR_4 \\ CR_5 \end{array} \begin{pmatrix} 1 & \frac{1}{5} & \frac{1}{3} & 0 & \frac{1}{3} \\ 5 & 1 & \frac{1}{4} & 0 & \frac{1}{5} \\ 3 & 4 & 1 & 0 & 0 \\ 0 & 0 & 0 & 1 & \frac{1}{5} \\ 3 & 5 & 0 & 5 & 1 \end{pmatrix}$$

According to AHP, $W_{cr\_PC}(1,j) = \dfrac{\sum_{j=1}^{5} R_{11j}}{\sum_{i=1}^{5} \frac{\sum_{j=1}^{5} R_{11j}}{5}}$

$$W_{cr\_PC}$$
$$= \begin{pmatrix} 0.06 & 0.21 & 0.36 & 0.11 & 0.10 & 0 & 0.13 & 0.07 & 0.09 & 0.39 & 0.03 & 0 & 0.07 \\ 0.20 & 0.10 & 0.07 & 0.13 & 0.21 & 0 & 0.13 & 0.14 & 0.13 & 0.23 & 0.07 & 0 & 0.08 \\ 0.25 & 0.14 & 0.10 & 0.49 & 0.49 & 0 & 0.47 & 0.09 & 0.49 & 0.14 & 0.53 & 0 & 0.17 \\ 0.05 & 0.12 & 0.29 & 0.12 & 0.07 & 0.1 & 0.21 & 0.18 & 0.15 & 0.17 & 0.31 & 0.7 & 0.19 \\ 0.44 & 0.43 & 0.18 & 0.14 & 0.13 & 0.9 & 0.06 & 0.52 & 0.14 & 0.05 & 0.06 & 0.3 & 0.49 \end{pmatrix}$$

$w^{(1)} = w_s \times W_{cr\text{-}PC} = (0.3380, 0.1628, 0.1909, 0.2353, 0.2469, 0.0550, 0.2399, 0.1272, 0.2325, 0.2335, 0.2245, 0.0850, 0.1365)$.

Gaining Mutual Importance among *PC* Elements

Using the Analytic Hierarchy Process to obtain mutual importance vector $w^{(2)}$ among elements in *PC*. Relative importance matrix $R'_i$ among $PC_i$ is evaluated by the expert team, and $R'_1$ between $PC_1$ and the others was taken as an example.

$$R'_1 = \begin{array}{c} \\ PC_1 \\ PC_2 \\ PC_3 \\ PC_4 \\ PC_5 \\ PC_6 \\ PC_7 \\ PC_8 \\ PC_9 \\ PC_{10} \\ PC_{11} \\ PC_{12} \\ PC_{13} \end{array} \begin{array}{c} \begin{array}{ccccccccccccc} PC_1 & PC_2 & PC_3 & PC_4 & PC_5 & PC_6 & PC_7 & PC_8 & PC_9 & PC_{10} & PC_{11} & PC_{12} & PC_{13} \end{array} \\ \left( \begin{array}{ccccccccccccc} 1 & \frac{1}{5} & \frac{1}{2} & \frac{1}{5} & \frac{1}{3} & 0 & 0 & 0 & \frac{1}{3} & \frac{1}{6} & 0 & \frac{1}{6} & \frac{1}{6} \\ 5 & 1 & \frac{1}{4} & \frac{1}{6} & \frac{1}{3} & \frac{1}{6} & 0 & \frac{1}{2} & 0 & \frac{1}{7} & 0 & \frac{1}{6} & 1 \\ 2 & 4 & 1 & \frac{1}{2} & 0 & \frac{1}{2} & 0 & 0 & 0 & \frac{1}{5} & 0 & 1 & \frac{1}{2} \\ 5 & 6 & 2 & 1 & \frac{1}{7} & \frac{1}{2} & 0 & \frac{1}{2} & \frac{1}{7} & \frac{1}{3} & \frac{1}{7} & 1 & \frac{1}{4} \\ 3 & 3 & 0 & 7 & 1 & 0 & \frac{1}{5} & 0 & \frac{1}{2} & \frac{1}{6} & 0 & \frac{1}{3} & \frac{1}{7} \\ 0 & 6 & 2 & 2 & 0 & 1 & 0 & \frac{1}{7} & \frac{1}{2} & \frac{1}{5} & 0 & 0 & \frac{1}{7} \\ 0 & 0 & 0 & 0 & 5 & 0 & 1 & 0 & \frac{1}{5} & \frac{1}{3} & 0 & \frac{1}{9} & \frac{1}{3} \\ 0 & 2 & 0 & 2 & 0 & 7 & 0 & 1 & \frac{1}{2} & \frac{1}{3} & 0 & 0 & \frac{1}{3} \\ 3 & 0 & 0 & 7 & 2 & 2 & 5 & 2 & 1 & \frac{1}{3} & \frac{1}{5} & 0 & \frac{1}{2} \\ 6 & 7 & 5 & 3 & 6 & 5 & 3 & 3 & 3 & 1 & \frac{1}{3} & \frac{1}{6} & \frac{1}{7} \\ 0 & 0 & 0 & 7 & 0 & 0 & 0 & 0 & 5 & 3 & 1 & \frac{1}{5} & 0 \\ 6 & 6 & 1 & 3 & 0 & 9 & 0 & 0 & 0 & 6 & 5 & 1 & 0 \\ 6 & 1 & 2 & 4 & 7 & 7 & 3 & 3 & 2 & 7 & 0 & 0 & 1 \end{array} \right) \end{array}$$

Similar to the calculation method of $W_{cr\text{-}PC}$, the first column vector of $W_{pc}$ was obtained: $Wpc(:,1) = (0.2359, 0.6712, 0.7462, 1.2086, 1.4282, 0.9495, 0.5368, 1.1111, 3.2800, 1.7718, 1.2462, 2.8462, 3.3077)$ according to $R_1$. Finally, similar to the calculation method of $w^{(1)}$, $w^{(2)} = (0.1110, 0.1321, 0.0852, 0.1231, 0.0742, 0.0173, 0.0952, 0.0952, 0.0903, 0.1123, 0.0548, 0.0100, 0.0300)$ was finally obtained.

Gaining *PC* Importance

Calculating the importance of *PC* according to Equation (5), $w = (0.1681, 0.0965, 0.0730, 0.1299, 0.0822 \ 0.0043, 0.1025, 0.0543, 0.0942, 0.1176, 0.0552, 0.0038, 0.0184)$.

### 4.1.3. Acquiring Functional-Unit = Importance

$W_i$ denotes the importance of functional unit i in the whole product concept, where $\sum_{i=1}^{n} W_i = 1$, and as for nine functional units of the hoist, ① lifting, ② balance, ③ stabilization, ④ working brake, ⑤ reducer, ⑥ support, ⑦ power transmission, ⑧ safety brake, and ⑨ master support, importance vector $W_i = (0.17, 0.11, 0.11, 0.10, 0.13, 0.09, 0.13, 0.07, 0.09)$ was obtained based on knowledge.

### 4.1.4. Obtaining *PC* Substructure Utility Vector

According to expert analysis, the impact of each candidate substructure on the *PC* was quantified by 0 to 9. As shown in Table 2, the larger the number is, the greater the impact. Particularly, 0 indicates that the substructure had no effect on this index.

**Table 2.** Substructure utility vector for a *PC*.

| | | | | | | | | | Substructure Utility Vector | | | | | | | | | |
|---|---|---|---|---|---|---|---|---|---|---|---|---|---|---|---|---|---|---|
| *PC* | s11 | s12 | s13 | s14 | s15 | s16 | s17 | s18 | s21 | s22 | s31 | s32 | s33 | s34 | S41 | ... | s91 | s92 |
| F1 | 7 | 7 | 8 | 8 | 9 | 9 | 9 | 9 | 2 | 2 | 5 | 7 | 9 | 1 | 5 | ... | 2 | 2 |
| F2 | 9 | 9 | 7 | 7 | 9 | 9 | 9 | 9 | 2 | 2 | 5 | 7 | 9 | 1 | 0 | ... | 2 | 2 |
| F3 | 9 | 9 | 9 | 9 | 7 | 7 | 7 | 7 | 5 | 3 | 5 | 9 | 7 | 1 | 3 | ... | 2 | 2 |
| F4 | 1 | 1 | 1 | 1 | 1 | 1 | 1 | 1 | 4 | 4 | 5 | 5 | 7 | 3 | 7 | ... | 2 | 2 |
| F5 | 5 | 5 | 5 | 5 | 7 | 7 | 7 | 7 | 4 | 4 | 5 | 5 | 7 | 3 | 7 | ... | 2 | 2 |
| F6 | 7 | 7 | 5 | 5 | 9 | 9 | 7 | 7 | 3 | 3 | 3 | 5 | 7 | 1 | 5 | ... | 2 | 2 |
| F7 | 0 | 0 | 0 | 0 | 0 | 0 | 0 | 0 | 0 | 0 | 0 | 0 | 0 | 0 | 9 | ... | 2 | 2 |
| F8 | 0 | 0 | 1 | 1 | 1 | 1 | 0 | 0 | 0 | 0 | 0 | 0 | 0 | 0 | 5 | ... | 0 | 0 |
| F9 | 7 | 7 | 5 | 5 | 7 | 7 | 9 | 9 | 6 | 6 | 7 | 7 | 7 | 3 | 0 | ... | 0 | 0 |
| F10 | 4 | 4 | 3 | 3 | 5 | 5 | 6 | 6 | 4 | 4 | 5 | 7 | 9 | 0 | 5 | ... | 0 | 0 |
| F11 | 0 | 0 | 0 | 0 | 0 | 0 | 0 | 0 | 0 | 0 | 0 | 0 | 0 | 0 | 0 | ... | 0 | 0 |
| F12 | 0 | 0 | 0 | 0 | 0 | 0 | 0 | 0 | 0 | 0 | 0 | 0 | 0 | 0 | 5 | ... | 7 | 9 |
| F13 | 3 | 3 | 3 | 3 | 3 | 3 | 3 | 3 | 3 | 3 | 3 | 3 | 3 | 7 | 7 | ... | 7 | 9 |

## 4.2. Model Solution Process

In order to solve the above model with the EGA, the functional units were mapped to the game players; the substructures were mapped to the strategy set for the player; and the constraints condition was mapped to the game rules. Perturbation probability pi of each functional unit was given according to the importance degree of the functional units for the entire hoist design, where $p_i$ = (0. 781, 0.547, 0.452, 0.343, 0.433, 0.536, 0.412, 0.246, 0.435), and maximum evolution generation $T$ = 250 was set according to multiple experiment simulations, as shown in Figure 9.

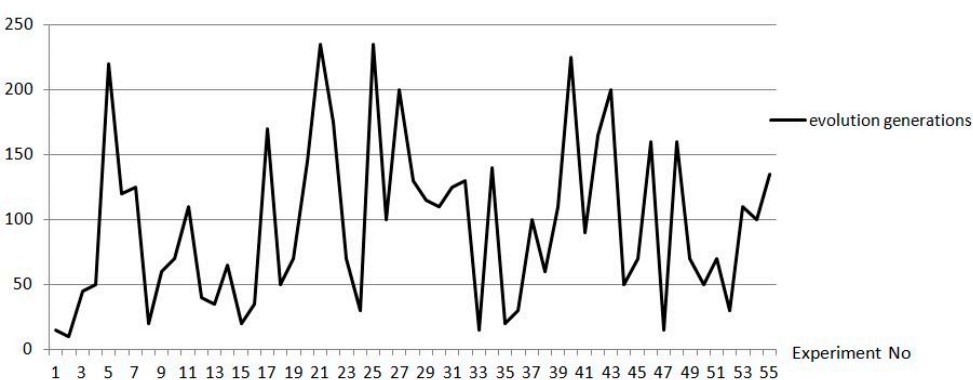

**Figure 9.** Nash equilibrium generation distribution.

Based on the above conditions, the evolutionary game was carried out in software environment MATLAB 8.3, in which the result of the conceptual design was shown in Figure 10. The optimal product design that meets the relevant *PC* was finally achieved as a single fold with center rope; fixed pulley placed vertically; horizontal speed reducer; balanced pulley placement; sliding bearing; wheel coupling; safety brake; and motor fixed pulley different side. The original data in this work were provided by a water-conservancy and hydroelectric-machinery company.

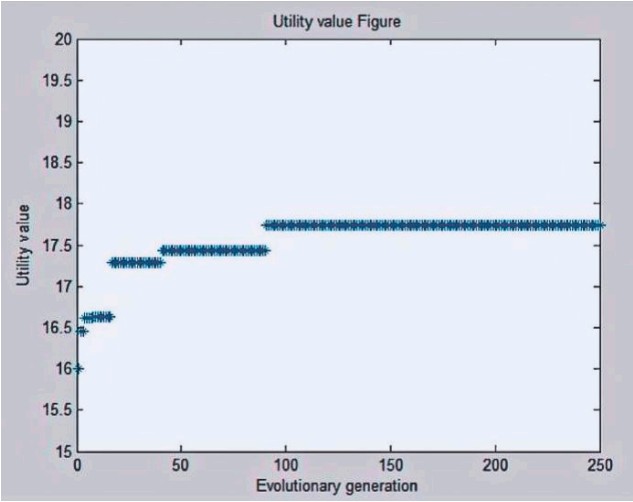

**Figure 10.** Solution optimization process.

*4.3. Results and Discussion*

Under the same conditions, the hoist concept was designed by the company's designers using an empirical design system of the company. As shown in Figure 11, the result was: single fold with center rope; balanced pulley placement; sliding bearing; wheel coupling; safety brake; and motor fixed pulley different side.

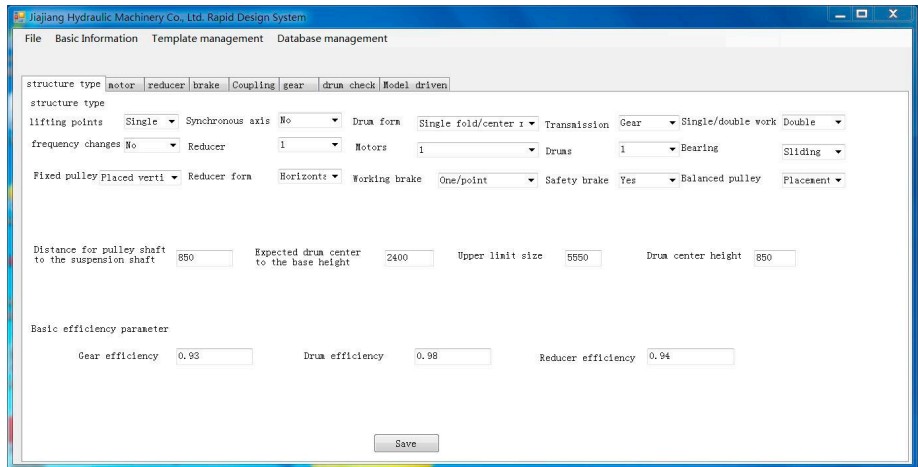

**Figure 11.** Current design-method result.

Comparing the concept designed by the company engineers with that achieved by the method proposed in this paper, the main difference was that engineers think that a motor fixed pulley different side makes the structure more compact, while there is neither CR nor PC related to compactness. From this point of view, the proposed design method in this paper is less advanced in the application of expert knowledge. A comprehensive comparison was made from occupation, design cycle, experiential knowledge, result reliability, and economy. Results are shown in Table 3.

Obviously, this method needs to be improved in the acquisition and learning of empirical knowledge, but performs well in other aspects.

**Table 3.** Performance comparison.

| Project | Occupation | Design Cycle | Experiential Knowledge | Result Reliability | Economy | Total |
|---|---|---|---|---|---|---|
| Current Design Method | ●Empirical design system ●Experienced designer | 1–2 days | ●Design experience ●Knowledge base | Reliable | Poor | General |
| Method of this Paper | Computer | 0.5 h | Improved knowledge base | Reliable | Well | great |
| Improved | Greatly | Greatly | Little | Little | Good | Greatly |

## 5. Conclusions

Product conceptual design was investigated in this paper. Based on the achieved experiment results, the following conclusions are derived:

It was found that a problem could be effectively solved by the method proposed in this paper. Using this process, the design cycle was reduced to 0.5 hour, and occupation and economy greatly improved.

The method does not perform well in empirical-knowledge application. Hence, our future work will focus on how to more accurately acquire design knowledge and objective–subjective expert knowledge.

**Author Contributions:** Conceptualization, Y.-L.H. and X.-B.H.; methodology, Y.-L.H.; software, Y.-L.H.; validation, X.-B.H., B.-Y.C. and R.-G.F.; formal analysis, X.-B.H.; investigation, Y.-L.H. and R.-G.F.; resources, R.-G.F.; data curation, R.-G.F. and Y.-L.H.; writing—original draft preparation, Y.-L.H. and B.-Y.C.; writing—review and editing, Y.-L.H.; visualization, Y.-L.H.; supervision, X.-B.H.; project administration, X.-B.H.; funding acquisition, X.-B.H.

**Funding:** This research was funded by Sichuan Province Science and Technology Support Program, grant number (2017GZ0146, 2018GZ0125); Made in China 2025-Sichuan Province program, grant number 2018ZZ011. The APC was funded by Sichuan Province Science and Technology Support Program.

**Conflicts of Interest:** The authors declare no conflict of interest.

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
