# Peer review of "A Product Conceptual Design Method Based on Evolutionary Game"

_machines, doi:10.3390/machines7010018_

Reviewer 1 Report

1. for the  case study: what is the exact value of parmmeter (pi , the perturbation probability and Termination condition τ)?

2. lack of comparison of results

Author Response

Dear reviewer

Thank you very much for your valuable review comments. Your professional review comments have greatly helped our work. We have corrected the relevant issues according to your guidance, and the following is the responses. We look forward to your guidance.

Point 1: for the case study: what is the exact value of parmmeter (pi , the perturbation probability and Termination condition T)?

Response 1: The exact values of pi and T are given in Section 4.2, where pi= (0. 781, 0.547, 0.452, 0.343, 0.433, 0.536, 0.412, 0.246, 0.435), and T=250 (Line 411).

Point 2: lack of comparison of results

Response 2: We added a result and discussion section to the paper to compare the current design method with the method presented in this paper, and more detailed analysis in the article, I look forward to your guidance again.

Reviewer 2 Report

Dear Authors

the paper is quite good.

I would like to ask you, please, to improve the following:

1) research of design methods

2) description of design methods used and other similar

3) description of the results

4) link between results and conclusions

5) English Language

Author Response

Dear reviewer

 Thank you very much for your valuable review comments. Your professional review comments have greatly helped our work. We have corrected the relevant issues according to your guidance, and the following is the responses. We look forward to your guidance.

Point 1:  research of design methods

Response 1: In the introduction, we introduced the research background and current status of the design method. Is it not enough? Please forgive me. I don’t understand the specific meaning of this question, and we look forward to your further comments.

Point 2: description of design methods used and other similar

Response 2: We added a result and discussion section to the paper to compare the current design method with the method presented in this paper. The main contents of the increase are shown in attachment

Point 3: description of the results

Response 3: In Section 4.3 of the paper, the results obtained using the current design method and the results obtained using the design method of this paper were compared and discussed. Including factors that cause differences in design results (Empirical knowledge acquisition and application)

Point 4: link between results and conclusions

Response 4: We rewrote the conclusion. Combining the results with other methods, based on the comparison with other methods, the performance of the method summarized.

Point 5: English Language

Response 5: Thank you very much for your comments. We know that our English is limited. Although try our best in the revision, there are still many problems. In the following work, we will work hard to improve our English level.We are very grateful for your guidance during your busy schedule, and hope that you can give us valuable comments again

Detailed changes are included in the paper. If there are other questions, please feel free to enlighten me.

Reviewer 3 Report

The authors have presented a paper on the optimization of a product design, based on soft computing techniques. The paper is presented well and the methods used are interesting. However, the paper requires revisions before it can be accepted for publication.

First of all, the authors need to proofread their manuscript as many errors can be found within the text, e.g. metter-element and many more.

In the Introduction, the authors need to specifically describe what is the novelty of the method proposed in this paper and how they are going to implement the novelties they have in mind. Furthermore, the authors mention that "While above methods contribute greatly to the process of the internalization of conceptual 67 design, they are still not ideal in terms of model expression and solution speed". Could you please elaborate more and be more specific on what is the meaning of this sentence?

In section 2, a short note on the benefits of the proposed method could be helpful.

There are two sections 3 in the manuscript. Please, correct this.

The case study is interesting. However, there is no benchmarking with an alternative method in order to compare the results and the efficiency of the method. In this respect, there is little proof of the value of the proposed method.

Finally, the conclusions need to be rewritten in order to exhibit some important features and results of the method rather than a description.

Author Response

Dear reviewer

Thank you very much for your valuable review comments. Your professional review comments have greatly helped our work. We have corrected the relevant issues according to your guidance, and the following is the responses. We look forward to your guidance.

Point 1: First of all, the authors need to proofread their manuscript as many errors can be found within the text, e.g. metter-element and many more.

Response 1: Thank you very much for your professional review comments. We have proofread the manuscript. Please review it again. If you have any comments, please let us know.

Point 2: In the Introduction, the authors need to specifically describe what is the novelty of the method proposed in this paper, and how they are going to implement the novelties they have in mind. Furthermore, the authors mention that "While above methods contribute greatly to the process of the internalization of conceptual 67 design, they are still not ideal in terms of model expression and solution speed". Could you please elaborate more and be more specific on what is the meaning of this sentence?

Response 2: Based on your review, we added a description of the novelty of the method in the Introduction. The mention that "While above methods contribute greatly to the process of the internalization of conceptual 67 design, they are still not ideal in terms of model expression and solution speed". The meaning of the expression is a bit ambiguous, I have modified it, please review it again.

Point 3: In section 2, a short note on the benefits of the proposed method could be helpful.

Response 3: Thank you for giving such a professional advice. We've added some content based on your suggestions to give a brief introduction to the benefits of this method. The added content is as follows:

2.4 Benefits

● Considering of PCs and CRs comprehensively, making products perform well in terms of performance and personalization.

● Modular product functions, and strives to achieve optimal design of each functional module under constraints.

● Using the EGA that performs well in solving the optimal combination problem solves the model, and quickly obtains the optimal solution.

Point 4: There are two sections 3 in the manuscript. Please, correct this.

Response 4: Thank you for pointing out the chapter questions. I have corrected them. Please continue to review our paper so that it can meet academic standards.

Point 5: The case study is interesting. However, there is no benchmarking with an alternative method in order to compare the results and the efficiency of the method. In this respect, there is little proof of the value of the proposed method.

Response 5: In Section 4.3 of the paper, the results obtained using the current design method and the results obtained using the design method of this paper were compared and discussed. Including factors that cause differences in design results (Empirical knowledge acquisition and application)

Point 6: Finally, the conclusions need to be rewritten in order to exhibit some important features and results of the method rather than a description.

Response 6: We rewrote the conclusion. Combining the results with other methods, based on the comparison with other methods, the performance of the method summarized.

Detailed changes are included in the paper. If there are other questions, please feel free to enlighten me.

Round  2

Reviewer 3 Report

The authors have put little effort in addressing the reviewer's comments. English editing is still required. The novelty of the paper is not supported by comparison to other works. The required explanation on a sentence is still vague The benefits can apply to many methods and not the proposed one specifically. The case study benchmarking is very poor (the figure is in Chinese!).

Author Response

Dear reviewer

Thank you very much for your valuable review comments that make our work more rigorous and scientific.

(1) For English language problems, I am going to hand over the manuscript to a professional English editor for retouching. The editor has not completed due to some problems in the payment. I hope that professional English editors can meet your requirements.

(2) The article adds comparisons with other references to illustrate the novelty of this article (Lines 71 to 78).

(3) "While above methods contribute greatly to the process of the internalization of conceptual design, they are still not ideal in terms of model expression and solution speed". What I want to express is that the main focus of the previous method is to study the commonality of various problem models. So for the solution of a specific problem, its model expression and solution speed is not good. I originally wanted to use this sentence to highlight the focus of this article, but the explanation is not enough, so it caused your doubts, I am very sorry.

(4) Combining the work of this paper, the benefits of this method are re-improved;

●Focus on the functional variables and constraints of the model, the solution obtained is the optimal solution that satisfies the constraint.

● Considering of PCs and CRs comprehensively, making products perform well in terms of performance and personalization.

●Modular product functions as players in the EGA that performs well on combinatorial optimization problems, and quickly obtains the optimal solution.

(5) With the help of the designer of the design system, the display interface was modified to display in English instead of Chinese.

For the first time in an international publication, many rules and regulations still require your corrections. Thank you again for your review.

Round  3

Reviewer 3 Report

The authors have improved their paper based on the reviewers' comments. The paper can be published as is.